# Stability and Bandgap Engineering of In_1−*x*_Ga_*x*_Se Monolayer

**DOI:** 10.3390/nano12030515

**Published:** 2022-02-01

**Authors:** Mattia Salomone, Federico Raffone, Michele Re Fiorentin, Francesca Risplendi, Giancarlo Cicero

**Affiliations:** 1Dipartimento di Scienza Applicata e Tecnologia, Politecnico di Torino, 10129 Torino, Italy; francesca.risplendi@polito.it (F.R.); giancarlo.cicero@polito.it (G.C.); 2Center for Sustainable Future Technologies, Istituto Italiano di Tecnologia, 10144 Torino, Italy; michele.refiorentin@iit.it

**Keywords:** beyond graphene, InSe, 2D materials, cluster expansion, kinetic Monte Carlo

## Abstract

Bandgap engineering of semiconductor materials represents a crucial step for their employment in optoelectronics and photonics. It offers the opportunity to tailor their electronic and optical properties, increasing the degree of freedom in designing new devices and widening the range of their possible applications. Here, we report the bandgap engineering of a layered InSe monolayer, a superior electronic and optical material, by substituting In atoms with Ga atoms. We developed a theoretical understanding of $In_{1-x}Ga_{x}Se$ stability and electronic properties in its whole compositional range (x=0−1) through first-principles density functional theory calculations, the cluster expansion method, and kinetic Monte Carlo simulations. Our findings highlight the possibility of modulating the InGaSe bandgap by ≈0.41 eV and reveal that this compound is an excellent candidate to be employed in many optoelectronic and photonic devices.

## 1. Introduction

Photonic technologies represent a rapidly emerging field, which is considered one of the most promising directions for industrial and scientific innovation, thanks to the wide variety of possible applications, ranging from quantum cryptography and quantum computing to sensors for biomedical imaging and light-emitting diodes [1,2,3,4,5,6,7]. Compared to traditional bulk photonic materials such as GaAs and Si, two dimensional (2D) materials own naturally passivated surfaces without any dangling bond, allowing for their easy integration with photonic structures [8,9,10,11]. They offer the opportunity of creating vertical heterostructures without the “lattice mismatch” issue, because of weak interactions between different monolayers (van der Waals forces). In addition, most 2D structures interact strongly with light and can be used to cover a wide electromagnetic spectrum thanks to their diverse electronic properties, ranging from insulating hexagonal boron nitride (hBN) [8,12] and semiconducting transition metal dichalcogenides (TMDs) [13,14] to semi-metallic graphene [15].

In this regard, InSe is a layered III-IV chalcogenides material that in recent years has attracted interest due to its optical and electronic properties [16,17]. In its monolayer phase, it shares the promising characteristics listed so far for 2D materials, and, in addition, it shows an extremely high carrier mobility (≈10,000 ${\mathrm{cm}}^{2}V^{-1}s^{-1}$) [18]. For photonic applications, it is required to develop 2D materials with continuous tunable bandgaps, and doping is one of the best techniques that can be exploited for this purpose [19]. Substitutional defects in a pristine monolayer may lead to a change in the lattice structure of the host material, resulting in a wide range bandgap tunability [20,21]. In our recent work [22], we defected 2D InSe with different substitutional impurities to generate well-defined and sharp defect states within the bandgap of the material, and we showed the possibility of using these new energy levels to achieve efficient emissions of single photons [23,24]. Here, considering the excellent optoelectronic properties and potential photonic applications shared by InSe and GaSe [25,26,27], and taking into account that they are almost lattice-matched and possess thickness-dependent tunable bandgaps [28,29] and very similar band structures (Figure 1b), we report a study of InGaSe compounds. In particular, we demonstrate the possibility of modulating the bandgap of an InSe monolayer by substituting In with Ga atoms. In the first part of the work, we study the thermodynamic stability and lowest energy structures of $In_{1-x}Ga_{x}Se$ compounds by means of the cluster expansion (CE) method coupled with density functional theory (DFT) simulations. We then verify $In_{1-x}Ga_{x}Se$ kinetic stability and present the more energetically favored lattice configurations for different indium to gallium fractions with a kinetic Monte Carlo (KMC) approach. In the second part of the work, we study the electronic properties of the more stable structures for different Ga concentrations in terms of band diagrams and density of states (DOS).

## 2. Methods

The CE is a method used to calculate the total energy of a compound existing in different configurations. As such, it is often exploited to predict stability and concentration-dependent phase diagrams of alloys/compounds. The lattice of a binary compound can be thought as composed of sites, which can be occupied either by atom A or atom B. This aspect is modeled in CE by means of an occupational variable $\sigma_{I}$ which can assume the values ±1, depending on a the atomic species occupying the I-th site. The total energy of a certain compound configuration is then given by the summation of the energy contribution associated with clusters (or figures) of these occupational variables (singlet, pairs, triplets... up to the M-body cluster, where M is the total number of sites in the lattice). The energy contribution of each cluster is referred to as the specific energetic contribution (ECI), indicated by J. The general Hamiltonian of a compound existing in several possible configurations can be written as [30]
(1)$$H\left(\vec{\sigma }\right)=\sum_{f}m_{f}J_{f}\langle \prod_{I\in f^{\prime }}\sigma_{I}\rangle .$$

Here, $m_{f}$ is the figure multiplicity which takes into account degenerate clusters, and $\vec{\sigma }$ is the configuration vector, which describes all the possible arrangements of atom A and B. The summation is over all clusters, which can involve one site (for singlets), two sites (for pairs), and so forth up to M. Because of the extremely high number of possible clusters, the summation is truncated. It was demonstrated that even with a finite number of terms the configurational energy can be represented with good precision [31,32].

DFT simulations were used to extract the ECIs necessary to find the Hamiltonian of the system by calculating the energy of a small set of structures with known lattice configurations [33]. Equation (Equation 1) is then reversed and filled with the data from DFT so that the ECIs are derived. The CE Hamiltonian, obtained by the calculated ECIs, is then used to quickly predict the energy of all configurations in a concentration range, allowing also for the study of structures that are too computationally demanding for DFT. The degree of accuracy of the expansion is monitored by a cross validation score, $\chi_{VL}$, which evaluates the difference between predicted and known energies (calculated by DFT) for different sets of data. We assume that when its value is ≈0.008 eV, which is well below the one used in the literature [34], the expansion guarantees reliable results. CE was performed by means of the Alloy Theoretic Automated Toolkit (ATAT) software package [33] and DFT calculations with the Quantum Espresso package [35,36].

For DFT simulations, we exploited ultrasoft pseudopotentials [37] to describe the electron-ion interaction and the gradient corrected Perdew Burke Ernzerhof (PBE) functional [38] to describe the exchange-correlation effects. The Kohn-Sham wavefunctions were expanded in plane waves with a kinetic energy cutoff of 80 Ry, to ensure converged stress values during lattice parameters optimization. The Brillouin Zone of the primitive cell was sampled by employing a 6 × 6 × 1 Monkhorst-Pack grid [39] for monolayer unit cells and reduced accordingly as the supercell dimensions increased to ensure the same k-point density. All calculations were performed without considering the effect of spin-orbit coupling (SOC), since its inclusion does not strongly affect InSe and GaSe band structures [22,40,41].

The KMC approach was used to study the kinetic stability and the more favored lattice configurations of InGaSe for different gallium concentrations. The process was carried out by simulating a sequence of diffusive events, each of which was selected on the basis of its transition rate [31,42,43]. The latter were affected by the energetics of the system calculated by means of the ECIs obtained previously with the CE. The simulations were performed exploiting the Zacros software [44,45]. We used this code to model the diffusion of gallium defects through the compound structure via positional exchanges between Ga and In. The lattice was composed of two stacked layers (Se atoms were not active participants in the simulated processes, because Ga substitutes only In; therefore, they were not considered).

## 3. Results and Discussion

Figure 2 shows the phase diagram obtained through CE. On the *x* axis appears the Ga concentration in $In_{1-x}Ga_{x}Se$ (*x* = 0 corresponds to pure InSe, while x=1 to pure GaSe), on the *y* axis, instead, are reported the formation energies calculated as
$$E_{FORM}=E_{In_{1-x}Ga_{x}Se}-\left[\left(1-x\right)\mu_{In}+x\mu_{Ga}+\mu_{Se}\right]$$
where *x* is the Ga concentration, $E_{In_{1-x}Ga_{x}Se}$ is the total energy of the corresponding compound structure and $\mu_{In}$, $\mu_{Se}$, and $\mu_{Ga}$ are the chemical potentials of In, Se, and Ga in their stable bulk phases (I4/mmm tetragonal indium, P$3_{1}21$ trigonal selenium, and Cmce gallium). Blue dots indicate energies of structures calculated with DFT, while gray crosses represent the energies of the structures that were predicted through the cluster expansion. Black dots indicate, instead, the energies of the ground states associated with the two pure phases, and the black dashed line that connects them marks the ground-state energy for any In/Ga fraction.

Analyzing the phase diagram, we are able to identify, for each concentration of Ga, the more stable structure, i.e., the one with a formation energy closer to the predicted ground state for a specific compound composition.

We found that a cluster expansion of 18 figures was sufficient to achieve a cross-validation score $\chi_{VL}$ of ≈0.008 eV, which satisfied our precision requirements. In the expansion, we considered the energetic contribution of the background (i.e., the one associated with the pristine InSe monolayer), the singlet interaction (related to the interaction of a defect with itself), and pair interactions (nearest neighbors’ figures, shortened to NN for simplicity) up to a diameter of 16.3 Å, labeled from NN2 to NN16. All the interactions discussed in the following involve atoms belonging to the same atomic “plane” (see Figure 1a as a reference), while interactions between interplane pairs were associated with smaller ECIs; therefore, their contribution to the total energy of the system was minor.

The compound total energy was increased (lowered) by positive (negative) ECIs, if we are considering interactions between atoms of the same species. Therefore, it is more probable to observe atoms of the same type forming patterns reproducing the figures associated with negative ECIs. In Figure 3, we report the least (Figure 3a) and the most favored (Figure 3b) patterns associated with pairs of the same species. It is apparent that figures NN2, NN4, and NN6, which represent close neighbor couples, were observed with small probabilities, while figures NN12, NN14, and NN16 were more likely to occur. In the supporting information we report all the figures considered in the expansion (Appendix A) along with their respective ECIs (Appendix A).

Some of the predicted energies (Figure 2, gray crosses) were within the cross-validation score sensitivity (their energy difference with respect to the predicted ground state for that specific Ga concentration was smaller than $\chi_{VL}$). Considering also that the CE was performed using as reference the two pure phases of InSe and GaSe, the interface energy between the two material was not considered explicitly; instead, we studied the kinetic stability of the $In_{1-x}Ga_{x}Se$ structure by KMC simulations. We verified the possibility of creating mixed phases by studying the kinetic stability of the $In_{1-x}Ga_{x}Se$ compounds. The obtained results provided information about the equilibrium In/Ga arrangement within the lattice. Since we are interested in understanding whether the two pure phases tend to separate when mixed together, we used lattices containing already formed clusters as the initial structures. Figure 4a,c reports the results for two significant cases: a GaSe quantum dot (x=0.14, supercell area of ≈26 nm${}^{2}$) and a GaSe stripe (x=0.50, supercell area of ≈65 nm${}^{2}$).

Observing Figure 4b,d, respectively, it can be noticed that InSe and GaSe (well separated at the beginning of the simulations) tended to mix, forming a homogeneous compound. The initial structures presented an interface energy, which had a dominant effect, and led to cluster dissolution. This findings were supported by the analysis of ECIs obtained from the cluster expansion. Indeed, the formation of clusters was hindered because it would increase the total energy of the compound, according to the ECIs shown in Figure 3a, for the whole gallium concentration range. Nevertheless, one can distinguish a recurring pattern (Appendix A), which was more often observed for low Ga concentration. This arrangement almost reflects the periodic repetition of the most stable figures obtained by the CE, reported in Figure 3b.

Our findings highlight that when indium atoms were substituted with gallium atoms, the two pure phases tended to mix, forming homogeneous stable lattice configurations. This result shows that, for this specific system, the interface energy between the two pure phases (not considered by the CE) was relevant and cannot be neglected to uniquely identify the equilibrium structure, hence, a KMC approach is necessary.

Starting from the results provided by the CE, we chose the lowest energy InGaSe structures in the whole concentration range and performed an in depth analysis of the electronic properties of the material. DFT calculations showed that the bandgap changed monotonically with the gallium concentration: the higher the percentage of Ga, the wider the bandgap (Figure 5a). In particular, $E_{G}^{d}$ (direct, calculated in Γ) ranged from 1.47 eV (pure InSe) to 1.88 eV (pristine GaSe), while $E_{G}^{i}$ (indirect, calculated between the valence band maximum, VBM, and the conduction band minimum, CBm) varied from 1.39 eV to 1.78 eV (see the supporting information for more details). The CB and VB of InSe at the Γ point showed a strong contribution of In *s*-orbitals and Se $p_{z}$-orbitals, respectively, (in agreement with other works [46,47,48]). However, the topmost levels of the valence band were also influenced by *p* orbitals of indium. These results were confirmed for high concentrations of In, such as, the $In_{0.83}Ga_{0.17}Se$ compound, which was the structure with the lowest Ga percentage between those studied to monitor the bandgap behavior. In Figure 6a–c, we plotted the k-resolved projected DOS, to highlight the contribution of each orbital to the band diagram. Very similar bandgap behaviors were obtained in other studies for other compounds containing In and Ga, in particular for $In_{1-x}Ga_{x}P$ and $In_{1-x}Ga_{x}As$ [49].

Finally, we aligned the $In_{1-x}Ga_{x}Se$ band diagrams (using the vacuum level as common reference) to assess the possibility of creating heterostructures with useful band alignment and studied the optical properties of the compound (this analysis is reported in the supporting information). Heterostructures are crucial components in many electronic, photonic, and optoelectronic devices such as field effect transistors, solar cells, and sensors; hence, their possible implementation using InGaSe would certainly widen the range of possible applications for this compounds. In Figure 5b, in particular, we reported the comparison considering direct bandgaps. We observed a sequence of type I heterostructures between InSe and the other compounds. The valence band maxima of $In_{0.83}Ga_{0.17}Se$, $In_{0.50}Ga_{0.50}Se$ and $In_{0.17}Ga_{0.83}Se$ were very close in energy; therefore, heterojunctions involving only these materials were not considered. Knowing that $k_{B}T$ at room temperature was ≈0.025 eV, it was apparent that most of the highlighted bandgap discontinuities were greater than this value, ranging from ≈2 $k_{B}T$ ($\Delta E_{V}$ InSe/$In_{0.17}Ga_{0.83}Se$) to 22 $k_{B}T$ ($\Delta E_{C}$ InSe/GaSe). This fact implies that, in principle, it is possible to make heterojunctions featuring efficient charge separation. As an example, one may build a multijunction solar cell by stacking InGaSe layers at increasing Ga concentrations and obtain a structure able to efficiently harvest a good portion of the solar spectrum.

## 4. Conclusions

In conclusion, we predicted $In_{1-x}Ga_{x}Se$ structure and thermodynamic stability exploiting the CE method. In addition, we overcame the limitation of neglecting the interface energy between the two pure phases with a KMC approach, and we showed the possibility of creating kinetically stable compounds, which did not exhibit phase separation and were characterized by concentration dependent bandgaps. The tuning was very effective and, considering the two pure phases, the variations of the energy gap values were 0.41 eV (direct) and 0.39 eV (indirect). In addition, heterostructures made of InSe/GaSe and a generic $In_{1-x}Ga_{x}Se$ compound exhibited high energy band discontinuities and can be used to actuate either charge separation or carrier confinement. These findings point out that InSe is an excellent candidate to be used in optoelectronics and photonics, offering many degrees of freedom in designing new devices for a wide range of possible applications.

## Figures and Tables

**Figure 1 nanomaterials-12-00515-f001:**
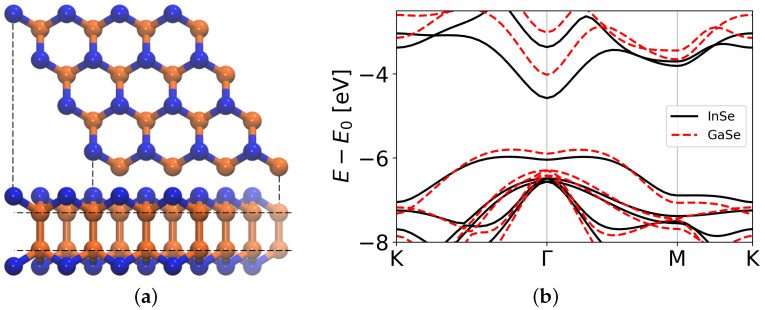
Top and side views (**a**) of the geometry of a pristine InSe (GaSe) monolayer. Selenium is represented in blue, while indium (gallium) is in orange. The monolayer is formed by four stacked atomic planes; therefore, two different levels in the In/Ga atoms arrangement can be distinguished (highlighted with the two horizontal dashed lines). This lattice arrangement is also shared by InGaSe compounds. (**b**) InSe band structure (black lines) and GaSe band structure (dashed red lines) calculated at the DFT-PBE level. In both cases, the vacuum level ($E_{0}$) is set to 0 eV.

**Figure 2 nanomaterials-12-00515-f002:**
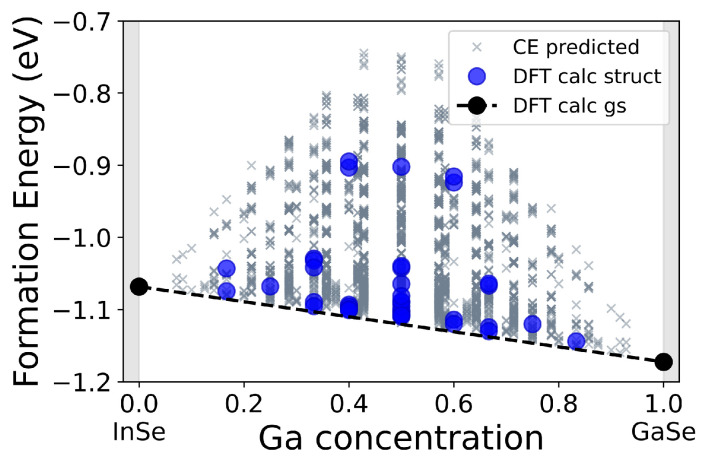
Phase diagram of the $In_{1-x}Ga_{x}Se$: gray crosses represent the energies of the structures predicted through the CE, blue dots indicate the energies found with DFT calculations performed on generated structures, black dots represent, instead, the energies of the two ground states associated with the two pure phases of InSe and GaSe, and the black dashed line marks the ground-state energy for all In/Ga fraction.

**Figure 3 nanomaterials-12-00515-f003:**
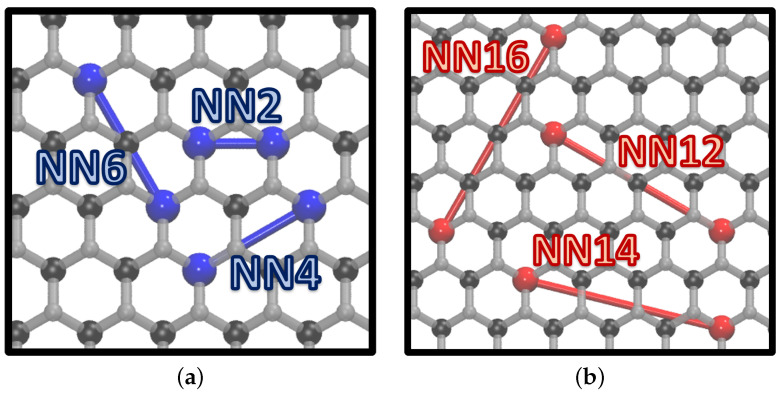
Representation of the less (**a**) and the more (**b**) stable pair figures obtained with the CE. ECIs related to red interactions (**b**) are negative; therefore, these configurations lower the total energy of the structure. Blue figures (**a**), instead, are associated with positive J coefficients, which implies that their formation is not energetically favored. Indium atoms are represented in black, selenium atoms in gray, and gallium atoms in blue (**a**) and red (**b**).

**Figure 4 nanomaterials-12-00515-f004:**
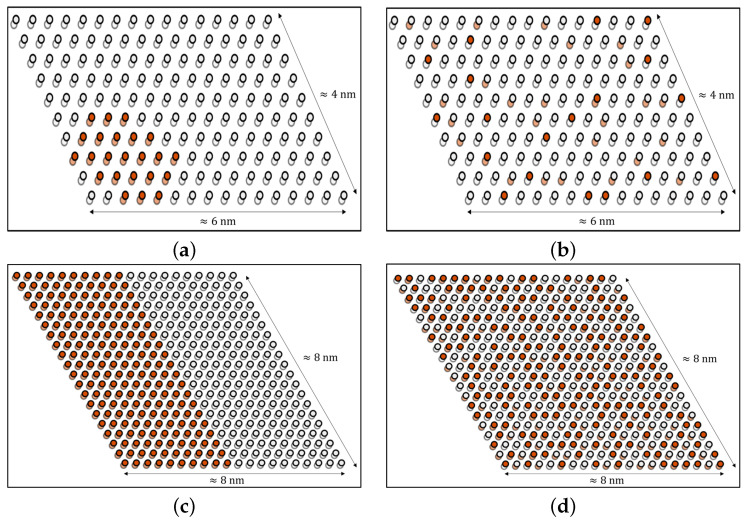
Panels (**a**,**c**) show the two initial KMC structures considered: a quantum dot and a stripe, respectively. Atoms belonging to the "bottom" plane are transparent, while their color represents the atomic species, gallium (red) and indium (light gray). Panels (**b**,**d**) report the equilibrium structures obtained from the two corresponding initial configurations shown in (**a**,**c**). It is apparent that in both cases the two pure phases tend to mix.

**Figure 5 nanomaterials-12-00515-f005:**
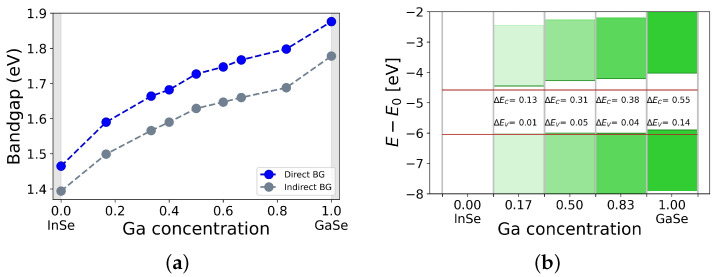
(**a**) $In_{1-x}Ga_{x}Se$ bandgap (direct blue, indirect gray) at varying Ga concentrations. (**b**) $E_{G}^{d}\left(x\right)$ alignment with respect to the vacuum level for some of the InGaSe structures.

**Figure 6 nanomaterials-12-00515-f006:**
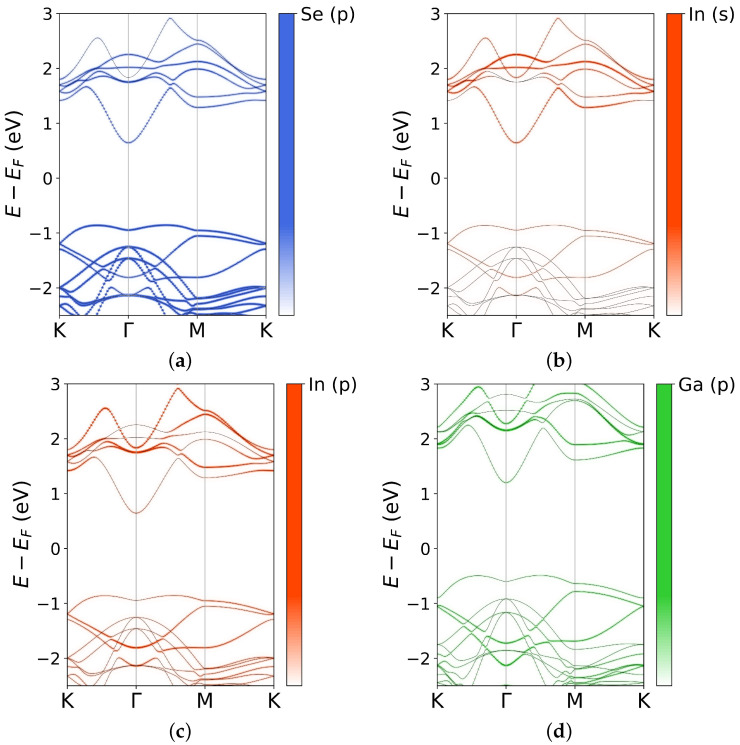
Band diagrams and k-resolved projected density of states, for $In_{0.83}Ga_{0.17}Se$ (**a**–**c**) and $In_{0.17}Ga_{0.83}Se$ (**d**). Selenium p orbitals provide a very similar contribution for both the considered compounds. *S* and *p* orbitals give a strong contribution both to conduction and valence bands, when the Ga percentage is low (panels **b**,**c**). When the Ga concentration is high, Ga *s* (not shown here) and *p* orbitals give a larger contribution.

## Data Availability

Not applicable.

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
