# Peer review of "Stability and Bandgap Engineering of In1−xGaxSe Monolayer"

_nanomaterials, 2022, doi:10.3390/nano12030515_

Round 1

Reviewer 1 Report

Authors used several calculation methods to predict the thermal stability and bandgap engineering of InSe by substituting Ga atoms. 

  1. Most part of the manuscript is talking about the stability, so the title shoudl inculde both stability and bandgap engineering.
  2. It seems to me that the bandgap of In1-xGaxSe is definitely between InSe and GaAs, in this case, what is the novety to calculate the bandgap with different Ga concentration. 
  3. Simarlar to the bandgap, is it that the thermal stability is also between InSe and GaSe? Experimently, the alloy will be formed with smallest energy automatically, is it that the structure configuration is certain?
  4. It will be more meaningful to calculate the optical anisotropy, carrier mobility and so on of InGaSe alloy 

Reviewer 2 Report

In this work, the authors report the bandgap engineering of layered InSe monolayer, a superior electronic and optical material, by substituting In atoms with Ga atoms. They developed a theoretical understanding of In1−xGaxSe stability and electronic properties in its whole compositional range  through first-principles Density Functional Theory calculations, Cluster Expansion method and Kinetic Monte Carlo simulations. This work will be of interest to other researchers in scientific and engineering community of 2D materials. At the same time, the reviewer has to point that in current manuscript there are concerns to be addressed. The concerns are:

1) The introduction has room to be further improved. In introduction, the authors write: “Photonic technologies represent a rapidly emerging field which is considered one of the most promising directions for industrial and scientific innovation, thanks to the wide variety of possible applications, ranging from quantum cryptography and quantum computing to sensors for biomedical imaging [1–4].” and “Compared to traditional bulk photonic materials such as GaAs and Si, two dimensional (2D) materials own naturally passivated surfaces without any dangling bond, allowing for their easy integration with photonic structures [5–8].” Photonic technologies have been widely applied in GaN-based light-emitting diodes (LEDs). It would be great if the authors include these new developments and achievements in the introduction, so to give the readers a much broader view. Several important references related to the application of photonic technologies in GaN-based LEDs, such as Nano Energy 69, 104427 (2020); Applied Physics Letters 118, 182102 (2021); Optics Express 27, A669 (2019), etc. should be added, so that the readers can be clear about the state-of-the-art of this topic.

2) Can you provide calculation equation of band gap for In1−xGaxSe alloy? What is the bowing parameter of the calculation equation of band gap for In1−xGaxSe alloy

3) The authors argue: “DFT calculations show that the bandgap changes monotonically with the gallium concentration: the higher the percentage of Ga, the wider is the bandgap.” Can you confirm the simulated band gap for In1−xGaxSe alloy using photoluminescence or electroluminescence measurements?

4) The authors claim: “Very similar bandgap behaviours were obtained in other studies for other compounds containing In and Ga, in particular for In1-xGaxP and In1-xGaxAs[46].” Were the similar bandgap behaviours obtained in III-nitride semiconductor such as InxGa1-xN alloy?

5) All references should follow the same formatting. Please check the style of references.

6) There are some grammatical errors in the manuscript, although most of them do not obscure the understanding of the technical contents. For example:

--“we study the electronic properties of the more stable structures for different Ga concentration” should be corrected to be “we study the electronic properties of the more stable structures for different Ga concentrations”

--“These results are confirmed for high concentrations of In, such as, for example, the In0.83Ga0.17Se compound,” should be corrected to be “These results are confirmed for high concentrations of In, such as, the In0.83Ga0.17Se compound,”

Round 2

Reviewer 1 Report

Authors have answerrd all my questions and it can be published already.